# Diagnostic performance and clinical implications of rapid SARS-CoV-2 antigen testing in Mexico using real-world nationwide COVID-19 registry data

Omar Yaxmehen Bello-Chavolla[1]*, Neftali Eduardo Antonio-Villa[2], Luisa Fernández-Chirino[3], Enrique C. Guerra[2], Carlos A. Fermín-Martínez[2], Alejandro Márquez-Salinas[2], Arsenio Vargas-Vázquez[2], Jessica Paola Bahena-López[2]

1 División de Investigación, Instituto Nacional de Geriatría, Mexico City, Mexico, 2 MD/PhD (PECEM), Faculty of Medicine, National Autonomous University of Mexico, Mexico City, Mexico, 3 Faculty of Chemistry, National Autonomous University of Mexico, Mexico City, Mexico

* oyaxbell@yahoo.com.mx

## Abstract

### Background

SARS-CoV-2 testing capacity is important to monitor epidemic dynamics and as a mitigation strategy. Given difficulties of large-scale quantitative reverse transcription polymerase chain reaction (qRT-PCR) implementation, rapid antigen tests (Rapid Ag-T) have been proposed as alternatives in settings like Mexico. Here, we evaluated diagnostic performance of Rapid Ag-T for SARS-CoV-2 infection and its associated clinical implications compared to qRT-PCR testing in Mexico.

### Methods

We analyzed data from the COVID-19 registry of the Mexican General Directorate of Epidemiology up to April 30th, 2021 (n = 6,632,938) and cases with both qRT-PCR and Rapid Ag-T (n = 216,388). We evaluated diagnostic performance using accuracy measures and assessed time-dependent changes in the Area Under the Receiver Operating Characteristic curve (AUROC). We also explored test discordances as predictors of hospitalization, intubation, severe COVID-19 and mortality.

### Results

Rapid Ag-T is primarily used in Mexico City. Rapid Ag-T have low sensitivity 37.6% (95%CI 36.6–38.7), high specificity 95.5% (95%CI 95.1–95.8) and acceptable positive 86.1% (95% CI 85.0–86.6) and negative predictive values 67.2% (95%CI 66.2–69.2). Rapid Ag-T has optimal diagnostic performance up to days 3 after symptom onset, and its performance is modified by testing location, comorbidity, and age. qRT-PCR (-) / Rapid Ag-T (+) cases had higher risk of adverse COVID-19 outcomes (HR 1.54 95% CI 1.41–1.68) and were older, qRT-PCR (+)/ Rapid Ag-T(-) cases had slightly higher risk or adverse outcomes and ≥7

**Data Availability Statement:** All data sources and R code are available for reproducibility of results at https://github.com/oyaxbell/covid_antigen_mx.

**Funding:** The publication of this article was supported by a grant from Secretaría de Educación, Ciencia, Tecnología e Innovación de la Ciudad de México CM-SECTEI/200/2020 "Red Colaborativa de Investigación Traslacional para el Envejecimiento Saludable de la Ciudad de México (RECITES)."

**Competing interests:** The authors have declared that no competing interests exist.

days from symptom onset (HR 1.53 95% CI 1.48–1.59). Cases detected with rapid Ag-T were younger, without comorbidities, and milder COVID-19 course.

## Conclusions

Rapid Ag-T could be used as an alternative to qRT-PCR for large scale SARS-CoV-2 testing in Mexico. Interpretation of Rapid Ag-T results should be done with caution to minimize the risk associated with false negative results.

## Introduction

SARS-CoV-2 testing capacity has been regarded as a fundamental factor to achieve pandemic control around the world [1]. It has been proposed that prompt isolation of possibly contagious individuals identified by testing and contact tracing is one of the most effective measures to reduce community-level transmission of SARS-CoV-2 infection; furthermore, effective reduction of community-level transmission can only be achieved with well-designed, universal, and cost-effective testing strategies [2]. Although quantitative reverse transcription polymerase chain reaction tests (qRT-PCR) have been the reference for detection of active SARS-CoV-2 infections, its systematic implementation entails significant technical difficulties in limited resource settings [3]. In order to address these methodological issues, the World Health Organization (WHO) proposed that rapid antigen tests (Rapid Ag-T) and other point-of-care tests (POCTs), which have demonstrated to have a high specificity compared to other molecular techniques [4, 5], could be useful alternatives for large-scale epidemiologic monitoring. With the worldwide rise in the use of Rapid Ag-T and POCTs, the presence of false negative results becomes of high epidemiologic importance, as unknown infected persons can be a vector of community transmission in countries where active SARS-CoV-2 infection is ongoing [6].

In Mexico, local authorities implemented a sentinel system-testing policy focused on tracking severe cases of COVID-19, and to a lesser extent those mild to moderate cases. Nevertheless, it has been reported that the implementation of full contact tracing procedure is only performed in areas where qRT-PCR testing facilities are available [7]. Rapid Ag-T have been recently promoted as a dynamic strategy for detection of active SARS-CoV-2 infection in Mexico to address these issues; however, despite being recommended by the WHO and used worldwide, few studies have evaluated their performance using large epidemiological real-world data [8, 9]. The increased use for Rapid Ag-T in Mexico demands a comprehensive evaluation for its diagnostic performance when compared with current reference testing techniques. Furthermore, its clinical implications could lead to the identification of subjects at risk for discrepancies of Rapid Ag-T results to minimize the risk of complications from COVID-19 and streamline prompt medical care. Here, we aim to assess the performance of Rapid Ag-T for diagnosis of SARS-CoV-2 infection and to examine the clinical implications of the discrepancies in its result compared to qRT-PCR test using national epidemiological dataset collected during the COVID-19 pandemic in Mexico.

## Methods

### Data sources

This is a retrospective analysis of the open COVID-19 registry dataset collected by the General Directorate of Epidemiology of the Mexican Ministry of Health within the National

Epidemiological Surveillance System (NESS), which includes daily updated suspected COVID-19 cases [10]. The database holds information on all persons tested for SARS-CoV-2 infection at public facilities in Mexico, as well as in all private healthcare facilities that follow the legal mandate to report COVID-19 cases to health authorities and the public locations for rapid antigen testing approved by the Mexican Ministry of Health. Furthermore, all samples registered in the NESS were conducted under the official guidelines from the Institute of Epidemiological Diagnosis and Reference (InDRE) to manage nasopharyngeal samples for rapid-antigen test [7]. This report adheres to the STARD guidelines for reporting of diagnostic accuracy tests [11]. A full list of available variables is presented in S1 File.

## Testing strategies for SARS-CoV-2 in Mexico

Prior to October 28[th], 2020, suspected cases were tested for SARS-CoV-2 infection using real-time qRT-PCR according to the Berlin Protocol [7]. Suspected COVID-19 cases were defined as an individual whom in the last 7 days has presented ≥2 of the following: cough, fever or headache; accompanied by either dyspnea, arthralgias, myalgias, sore throat, rhinorrhea, conjunctivitis or chest pain. Amongst suspected cases, the Ministry of Health established two protocols for case confirmation: 1) SARS-CoV-2 testing is done widespread for suspected COVID-19 cases with severe acute respiratory infection and signs of breathing difficulty or deaths with suspected COVID-19, 2) for all other cases, a sentinel surveillance model is being utilized, whereby 475 health facilities, which comprise a nationally representative sample, evaluate ~10% of mild outpatient cases to provide estimates of mild cases [7, 12].

After October 28[th], 2020, tests for SARS-CoV-2 infections additionally included those who were detected using one of the three available Rapid Ag-T including STANDARD™Q COVID-19 Ag Test, Panbio™ COVID-19 Ag RAPID Test Device, and Sofia2 SARS Antigen FIA by Quidel Corporation, which are approved to use and evaluated for efficacy by the National Institute for Epidemiological Diagnosis and Reference and the WHO [13]. These Rapid Ag-T are available in healthcare community-level locations for testing of suspected COVID-19 cases or subjects traced by epidemiological association with a suspected case and they are used extensively for monitoring and tracking COVID-19 incidence rates in Mexico City [14]. A full list of qRT-PCR and Rapid Ag-T kits available and approved for its use in Mexico is presented in **S1 File**. Confirmed SARS-CoV-2 infection is defined as an individual with a positive Rapid Ag-T or a positive qRT-PCR test. Cases with negative Rapid Ag-T but with close contact with a confirmed SARS-CoV-2 case and/or compatible clinical symptoms of COVID-19 were eligible for further evaluation with qRT-PCR testing within testing facilities [7]. According to InDRE, all negative cases need be re-tested at the same moment with qRT-PCR test to confirm or discard the diagnosis of SARS-CoV-2 infection.

## Definitions of outcomes and predictors

For cases who had both qRT-PCR and rapid antigen test information available, we used the qRT-PCR result as a reference test to classify cases as true positive (qRT-PCR + / Rapid Ag-T +), true negative (qRT-PCR—/ Rapid Ag-T -), false positive (qRT-PCR—/ Rapid Ag-T +) and false negative (qRT-PCR + / Rapid Ag-T -). Severe outcomes were defined as a composite of either death, ICU admission or requirement for invasive ventilation; hospital admission, requirement for intubation and lethality were also evaluated as outcomes. Follow-up time was estimated in days from symptom onset until either hospitalization or death, depending on the outcome of interest, or censoring, whichever occurred first.

## Statistical analysis

**Population-based statistics.**   We compared testing rates standardized per 100,000 inhabitants across Mexican municipalities and its trends over time after its implementation in late October comparing testing rates between Mexico City and the rest of Mexico due to the high density of Rapid Ag-T in the former. We also compared incident cases detected with qRT-PCR and Rapid Ag-T in Mexico City compared to the rest with the country. Furthermore, we compared cases who were assessed exclusively with Rapid Ag-T, qRT-PCR, or both to identify factors which influence testing in these settings.

**Performance of rapid antigen tests compared to qRT-PCR.**   We evaluated the performance of Rapid Ag-T compared to qRT-PCR using complete-case analysis of individuals who had both results available using confusion matrices and areas under the receiving operating characteristic curves (AUROC) with the *caret* and *pROC* R packages. We estimated the concordance of both testing methods using *Cohen's* Kappa (κ) coefficient. We further estimated sensitivity, specificity, positive and negative predictive values (PPV, NPV, respectively) and positive and negative likelihood ratios (LR+ and LR-, respectively) and their corresponding 95% confidence intervals with DeLong's method with the *OptimalCutpoints* R package. To evaluate the performance of Rapid Ag-T in different settings, we stratified these metrics according to testing location (Mexico City vs. Rest of Mexico), patient status (inpatient vs. outpatient), cases with and without comorbidities, age (>60 vs. ≤60 years) and time from symptom onset to evaluation (>7 vs ≤7 days from onset).

**Time-dependent performance of rapid antigen tests.**   We evaluated time-varying diagnostic performance or Rapid Ag-T using time-dependent ROC curves with the *timeROC* R package with inverse probability weighting in Cox regression, adjusted for age and sex for 1, 3, 5, 7, 10 and 15 days from symptom onset. We further evaluated the performance of Rapid Ag-T to predict hospitalization, mortality, and intubation, in cases with or without added qRT-PCR testing using the same proposed cut-offs.

**Predictors of test discordances.**   We tested for predictors of Rapid Ag-T and qRT-PCR discrepancies using mixed effects logistic regression, considering heterogeneity in epidemic dynamics across Mexico including municipality of residence as a random effect. To dissect predictors of test discrepancy, for false positive models we included only false positive and true negative cases and for false negative models we included true positive and false negative cases. We adjusted all models for age, sex, time from symptom onset and number of comorbidities.

**Clinical implications of Rapid Ag-T results.**   We evaluated test discordances as predictors for hospitalization, lethality and the composite event of severe outcomes using mixed-effects Cox regression incorporating municipality of residence as a random effect within the frailty term to control for geographical heterogeneity. Both for outcome predictors and for the implication of test discordances on intubation rates, we fitted mixed effects logistic regression models adjusted for age, sex, diabetes, arterial hypertension, CORP, immunosuppression, cardiovascular disease, obesity, asthma, and chronic kidney disease and considering municipality of residence as a random intercept. Next, we evaluated predictors for hospitalization, lethality and the composite event of severe outcomes using mixed-effects Cox regression in SARS--CoV-2 cases detected using Rapid Ag-T. For Cox models, proportional risk assumptions were verified using Schönfeld residuals and visual inspection of time-varying effects; for logistic regression models, goodness of fit was evaluated using the Hosmer-Lemeshow test and model selection was carried out using minimization of the Bayesian Information Criterion (BIC). All statistical analyses were conducted using R language version 4.0.3 and a p-value <0.05 was considered as the statistical significance threshold.

## Results

### Study population

Until April 30, 2021 a total of 6,307,964 subjects had been tested for SARS-CoV-2 in Mexico. Amongst them, 3,703,978 (58.7%) had only qRT-PCR test, 2,387,598 (37.8%) had only a rapid antigen test and 216,388 (3.4%) subjects had both qRT-PCR and rapid antigen tests. When comparing characteristics amongst the three previous groups, cases tested using Rapid Ag-T were younger, predominantly female, had lower rates of chronic comorbidities, and fewer cases who presented with features of severe COVID-19. Notably, cases who undertook Rapid Ag-T had lower median days from symptom onset to clinical assessment (Table 1).

Amongst tested cases, a total of 2,194,872 (34.7%) cases had confirmed SARS-CoV-2 infection using either of those tests and 4,133,092 (65.4%) had a negative result. The positivity rate was lower for Rapid Ag-T (22.1%) compared to qRT-PCR (43.2%). After the implementation of Rapid Ag-T, the rate of testing using this method was the largest in Mexico City, with over 18,767.32 tests per 100,000 habitants, followed by the state of Morelos with 5,619.06 Rapid Ag-T per 100,000 habitants and Queretaro with 2,004.04 Rapid Ag-T per 100,000 habitants. Between October 2020 and April 2021, ~80% of SARS-CoV-2 testing in Mexico City was carried out using Rapid Ag-T. compared with <10% in the rest of Mexico. Nevertheless, since January up to April 2021 there was a rapid increase of Rapid Ag-T across Mexico. Overall, amongst 527,068 confirmed SARS-CoV-2 cases using Rapid Ag-T, 296,823 (56.3%) were confirmed in Mexico City (Fig 1). A STARD diagram depicting all evaluated cases and a histogram of time from onset to testing is presented in S1 File.

**Table 1. Characteristics of subjects tested for SARS-CoV-2 infection in Mexico, comparing cases who were tested using qRT-PCR, rapid antigen tests and a combination of both.**

| Parameters | RT-PCR test n = 3,703,978 | Rapid Ag test n = 2,387,598 | Both tests n = 216,388 | p-value |
|---|---|---|---|---|
| Age (years) | 41.8 (±17.2) | 39.3 (±16.0) | 43.9 (±18.6) | <0.001 |
| Male sex (%) | 1,792,679 (48.4) | 1,136,693 (47.6) | 101,943 (47.1) | <0.001 |
| Confirmed SARS-CoV-2 (%) | 1,599,793 (43.2) | 527,068 (22.1) | 68,011 (31.4) | <0.001 |
| Diabetes (%) | 427,101 (11.5) | 165,722 (6.9) | 30,983 (14.3) | <0.001 |
| COPD (%) | 41,757 (1.1) | 10,985 (0.5) | 3,642 (1.7) | <0.001 |
| Asthma (%) | 95,879 (2.6) | 45,175 (1.9) | 5,566 (2.6) | <0.001 |
| Immunosuppression (%) | 38,012 (1) | 9,487 (0.4) | 2,648 (1.2) | <0.001 |
| Hypertension (%) | 574,606 (15.5) | 232,620 (9.7) | 40,870 (18.9) | <0.001 |
| Other (%) | 76,987 (2.1) | 21,892 (0.9) | 7,086 (3.3) | <0.001 |
| CVD (%) | 61,951 (1.7) | 17,344 (0.7) | 5,006 (2.3) | <0.001 |
| Obesity (%) | 501,168 (13.5) | 198,742 (8.3) | 26,467 (12.2) | <0.001 |
| CKD (%) | 58,442 (1.6) | 13,335 (0.6) | 6,365 (2.9) | <0.001 |
| Smoking (%) | 291,310 (7.9) | 206,116 (8.6) | 20,083 (9.3) | <0.001 |
| Pneumonia (%) | 411,995 (11.1) | 37,307 (1.6) | 31,877 (14.7) | <0.001 |
| Hospitalization (%) | 590,261 (15.9) | 49,643 (2.1) | 56,660 (26.2) | <0.001 |
| ICU admission (%) | 532,797 (14.4) | 48,205 (2) | 55,131 (25.5) | <0.001 |
| Intubation (%) | 222,698 (1.9) | 3,425 (0.1) | 2,818 (1.3) | <0.001 |
| Death (%) | 222,698 (6) | 20,469 (0.9) | 18,603 (8.6) | <0.001 |
| Time to assessment* (days) | 3 (1–5) | 2 (0–4) | 3 (1–5) | <0.001 |

**Abbreviations:** qRT-PCR: Reverse transcription polymerase chain reaction; CKD, Chronic Kidney Disease; CVD: cardiovascular disease; COPD: Chronic Obstructive Pulmonary Disease; OR: Odds Ratio; 95%CI: 95% Confidence interval, ICU: Intensive Care Unit. *Footnotes*: Age is presented as mean and standard deviation. Time to assessment is presented in median and interquartile range. *Time to assessment is defined as the time since COVID-19 related symptoms onset up to the registration of the tested subject in the medical unit. Global comparison of all three groups using ANOVA, Kruskal-Wallis, or Chi-Square teste wherever appropriate.

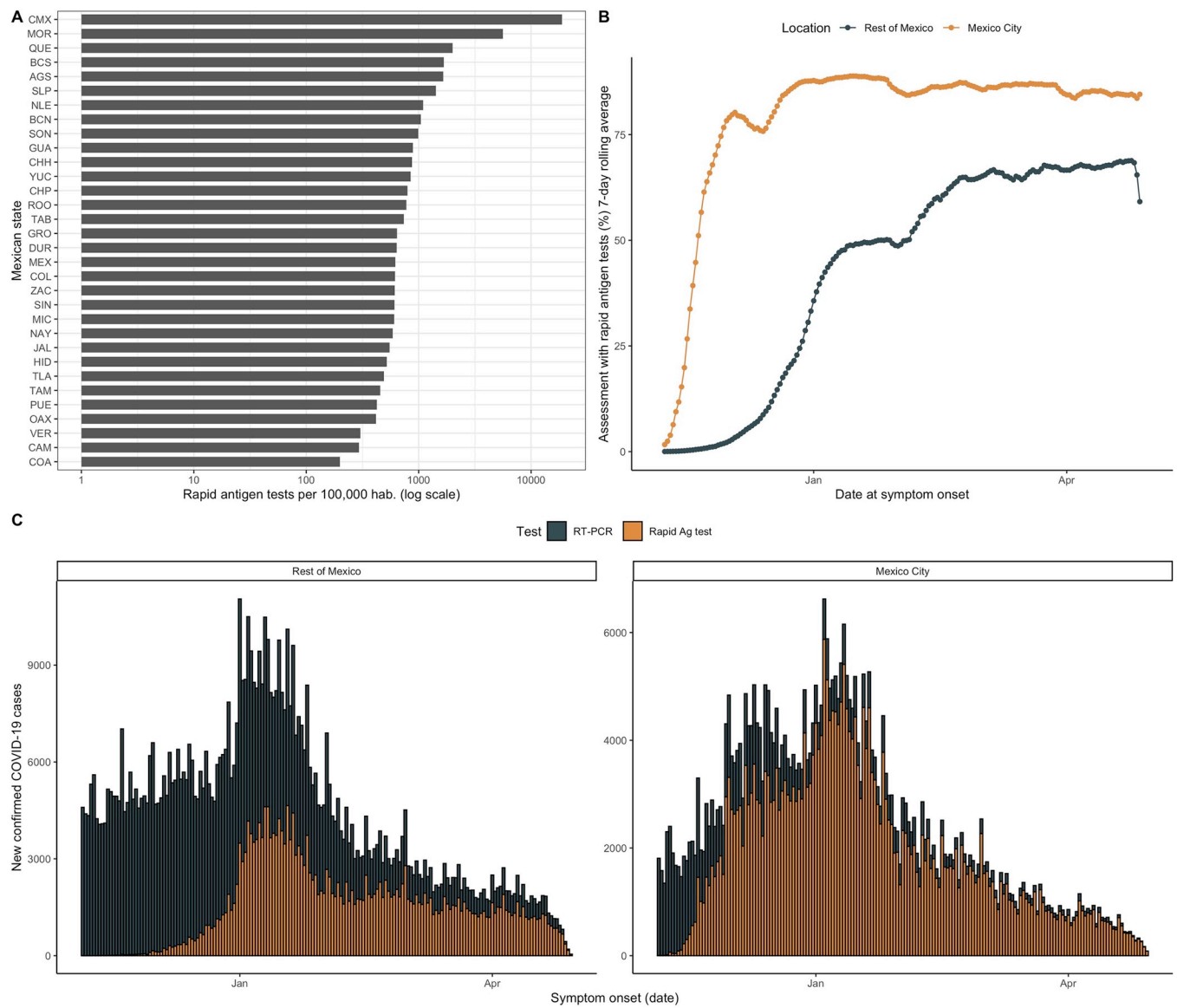

**Fig 1.** A) Number of rapid antigen tests per 100,000 population across different Mexican states. Figure also shows the percentage of rapid antigen tests amongst all SARS-CoV-2 tests administered in Mexico City and the rest of Mexico (B) and the curve of confirmed cases according to date from symptom onset in Mexico City and the rest of Mexico (C, D). **Abbreviations:** qRT-PCR: Reverse transcription polymerase chain reaction.

**Performance of rapid antigen tests compared to qRT-PCR.** A total of 216,388 subjects were tested using both qRT-PCR and Rapid Ag-T. Amongst them, 18,373 had pending tests results and 4,191 had inadequate qRT-PCR samples. Overall, a total of 193,824 cases had valid qRT-PCR and rapid antigen test results (**S1 File**). Subjects tested with qRT-PCR test had higher rates of chronic comorbidities and COVID-19 complications compared to cases with Rapid Ag-T only, but lower rates compared to qRT-PCR only (**Table 1**). Overall, we observed low concordance between both test modalities (κ = 0.368, 95%CI 0.363–0.372); when considering qRT-PCR as reference test, we identified 20,738 (10.69%) true positives, 128,464 (21.17%) false negatives, 3,572 (1.84%) false positives and 41,050 (66.2%) true negatives, yielding and AUROC of 0.666 (95%CI 0.660–0.671). Overall, we identified that rapid antigen tests

**Table 2. Overall diagnostic performance metrics of rapid antigen tests to detect SARS-CoV-2 infection compared to qRT-PCR in Mexico and stratification by region of testing, patient setting, comorbidity, age and days from symptom onset to evaluation.**

| Parameter | FP/FN | AUROC (95%CI) | Sensitivity (%, 95%CI) | Specificity (%, 95%CI) | PPV (%, 95%CI) | NPV (%, 95%CI) | LR+ (95%CI) | LR- (95%CI) |
|---|---|---|---|---|---|---|---|---|
| Overall | 479/ 4917 | 0.666 (0.66–0.671) | 37.6 (36.6–38.7) | 95.5 (95.1–95.9) | 86.1 (85–86.6) | 67.2 (66.2–69.2) | 8.3 (7.6–9.1) | 0.65 (0.64–0.66) |
| Mexico City | 255/ 4108 | 0.658 (0.652–0.664) | 34.5 (33.3–35.6) | 97.1 (96.8–97.5) | 89.4 (88.2–89.9) | 67.8 (66.6–70.5) | 12.0 (10.6–13.6) | 0.67 (0.66–0.69) |
| Rest of Mexico | 224/809 | 0.683 (0.668–0.698) | 50.0 (47.5–52.5) | 86.6 (84.9–88.2) | 78.3 (75.8–79.9) | 64.2 (61.9–67.5) | 3.7 (3.3–4.3) | 0.58 (0.55–0.61) |
| Outpatient | 370/ 4195 | 0.652 (0.646–0.658) | 34.2 (33.1–35.4) | 96.2 (95.8–96.6) | 85.5 (84.2–86.2) | 69.1 (68–71.4) | 9.0 (8.1–10.0) | 0.68 (0.67–0.70) |
| Inpatient | 109/722 | 0.692 (0.674–0.709) | 52.0 (49.5–54.6) | 86.3 (83.8–88.6) | 87.8 (85.4–88.8) | 48.8 (46.3–54.1) | 3.8 (3.2–4.6) | 0.56 (0.52–0.59) |
| No comorbidities | 294/ 3092 | 0.660 (0.653–0.667) | 36.3 (35–37.7) | 95.7 (95.2–96.2) | 85.7 (84.2–86.4) | 67.8 (66.5–70.4) | 8.4 (7.5–9.5) | 0.67 (0.65–0.68) |
| ≥1 comorbidity | 182/ 1811 | 0.674 (0.665–0.684) | 39.7 (38–41.5) | 95.1 (94.4–95.8) | 86.8 (85–87.6) | 66.1 (64.5–69.5) | 8.1 (7.0–9.4) | 0.63 (0.61–0.65) |
| <60 years | 392/ 4164 | 0.657 (0.651–0.663) | 35.6 (34.4–36.8) | 95.8 (95.4–96.2) | 85.4 (84.1–86.1) | 68.3 (67.2–70.5) | 8.5 (7.7–9.4) | 0.67 (0.66–0.69) |
| ≥60 years | 87/753 | 0.699 (0.684–0.714) | 47.0 (44.4–49.7) | 92.7 (91.1–94.1) | 88.5 (86.1–89.5) | 59.6 (57–65) | 6.5 (5.2–8) | 0.57 (0.54–0.6) |
| <7d from onset | 432/ 4050 | 0.674 (0.668–0.68) | 39.3 (38.1–40.5) | 95.5 (95.1–95.9) | 85.8 (84.6–86.4) | 69.5 (68.4–71.5) | 8.8 (8–9.7) | 0.64 (0.62–0.65) |
| ≥7d from onset | 47/867 | 0.618 (0.603–0.632) | 28.6 (26.1–31.3) | 94.9 (93.2–96.2) | 88.1 (84.7–89.4) | 50.1 (46.9–57.9) | 5.6 (4.2–7.5) | 0.75 (0.72–0.78) |

**Abbreviations:** FP: False positive; FN: False negative; PPV: Positive Predictive Value; NPV: Negative Predictive Value, LR+: Positive Likelihood Ratio, LR-: Negative Likelihood Ratio, qRT-PCR: Reverse transcription polymerase chain reaction; AUROC: Area under the receiving operating characteristic curve.

have a sensitivity of 37.6% (95%CI 36.6–38.7) and a specificity of 95.5% (95%CI 95.1–95.9), with a PPV of 86.1% (95%CI 85.0–86.6), a NPV of 67.2% (95%CI 66.2–69.2), a LR+ of 8.3 (95%CI 7.6–9.1), and a LR- of 0.65 (95%CI 0.64–0.66). Next, we assessed how the Rapid Ag-T performed in different scenarios and identified a lower performance in Mexico City compared to the rest of Mexico, for outpatients, younger cases, cases without comorbidities and, notably, in cases who had ≥7 days from symptom onset at evaluation (**Table 2**).

**Time-dependent variation in Rapid Ag-T performance.** As a sensitivity analysis, we used time-dependent ROC curves to model changes in diagnostic performance over time for detection of SARS-CoV-2 infection. Notably, Rapid Ag-T had unreliable test performance when disaggregating by time from symptom onset, which could be related to variability in CT values over the course of SARS-CoV-2 infections [15]. Here, we observed that, compared to qRT-PCR, Rapid Ag-T had the better AUROC at 3 days and its performance subsequently decreased until reaching the lowest AUROC at 15 days after symptom onset, adjusted for age and sex. Next, we evaluated whether using one test modality provided better predictive capacity for hospitalization, intubation and mortality, which would have relevant clinical implications for test selection. We observed that a positive qRT-PCR was relatively better at predicting hospital admission between 10–15 days after symptom onset and mortality at days 10–15, with a poor utility for all outcomes. For Rapid Ag-T, we observed similar trends, with a window for hospitalization and mortality between days 7–10 after symptom onset, but very low overall time-dependent test performance (**Table 3**).

**Table 3. Time-dependent area under ROC curves using inverse weighted probability with Cox regression for detection of SARS-CoV-2 infection, adjusted for age assessing the performance of rapid antigen tests compared to qRT-PCR at days 1, 3, 7, 10 and 15.** The table also shows the ability of qRT-PCR or rapid antigen tests to predict hospitalization, intubation and mortality related to COVID-19 at these different time points.

| Time AUROC | qRT-PCR vs. Ag-T | qRT-PCR Hospitalization | Ag-T Hospitalization | qRT-PCR intubation | Ag-T Intubation | qRT-PCR Mortality | Ag-T Mortality |
|---|---|---|---|---|---|---|---|
| 1 day | 0.564 | 0.455 | 0.484 | 0.36 | 0.451 | 0.568 | 0.555 |
| 3 days | 0.582 | 0.464 | 0.492 | 0.378 | 0.465 | 0.571 | 0.558 |
| 7 days | 0.572 | 0.521 | 0.52 | 0.463 | 0.503 | 0.632 | 0.584 |
| 10 days | 0.563 | 0.567 | 0.563 | 0.553 | 0.544 | 0.676 | 0.591 |
| 15 days | 0.556 | 0.604 | 0.556 | 0.635 | 0.545 | 0.692 | 0.572 |

**Abbreviations:** AUROC: Area under the receiving operating characteristic curve; qRT-PCR: Reverse transcription polymerase chain reaction; Ag-T: Rapid Antigen test.

**Clinical characterization of cases with discordant Rapid Ag-T results.** We evaluated predictors of false positive and false negative results in Rapid Ag-T for SARS-CoV-2 using qRT-PCR as reference test. Cases with false negative results had ≥7 days from symptom onset, were younger, and predominantly female. Regarding comorbidities, cases with false negative results were less likely to have underlying immunosuppression, obesity and chronic kidney disease. Regarding false positive results, we only observed increasing age as a significant predictor, with a non-significant trend in cases with chronic kidney disease (**Fig 2**). Next, we investigated whether these test discordances were predictive of COVID-19 outcomes. Regarding hospitalization we observed that, compared to true negative results, risk for hospitalization was higher for cases with true positive (HR 1.05, 95%CI 1.03–1.08) Rapid Ag-T results when compared with true negatives, adjusting for treatment setting, comorbidities, sex and age. Compared to true negative results, risk of intubation requirement was higher for false positive test results (OR 2.21, 95%CI 1.78–2.75), followed by true positive (OR 1.89, 95%CI 1.69–2.11)

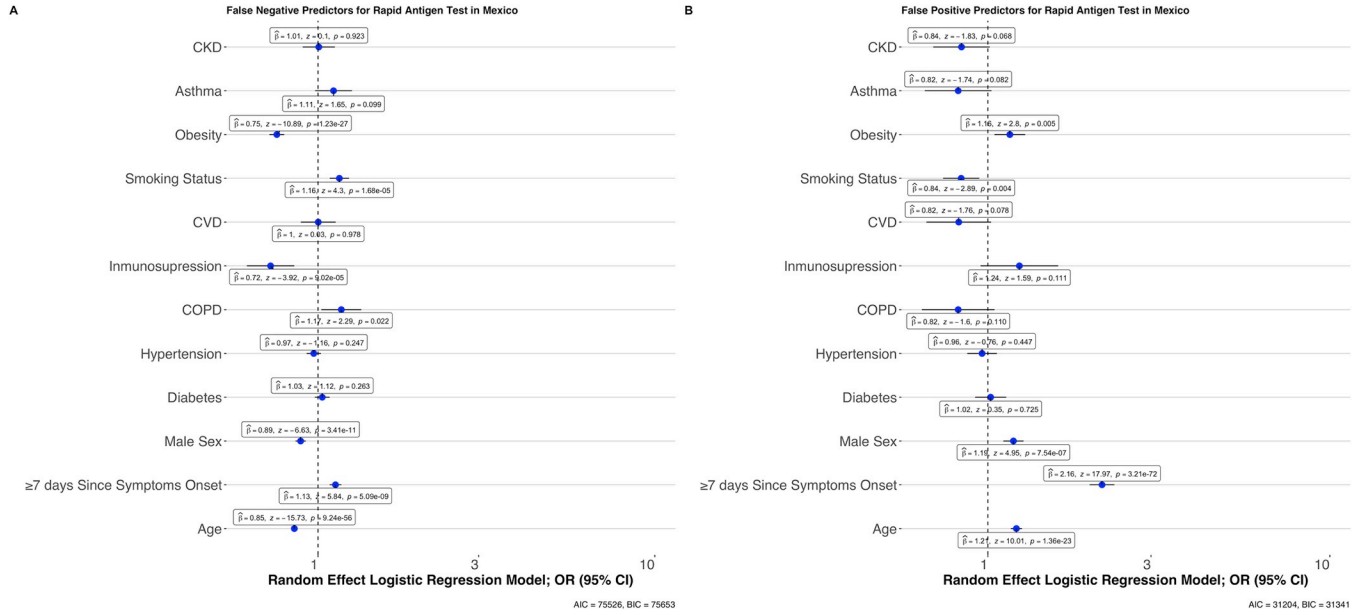

**Fig 2.** Mixed effects logistic regression models assessing predictors of cases with false negative compared to true positive test results (A) and false positive compared to true negative test results (B) using qRT-PCR as reference tests. **Abbreviations:** qRT-PCR: Reverse transcription polymerase chain reaction; CKD, Chronic Kidney Disease; CVD: cardiovascular disease; COPD: Chronic Obstructive Pulmonary Disease; OR: Odds Ratio; 95%CI: 95% Confidence interval, ICU: Intensive Care Unit.

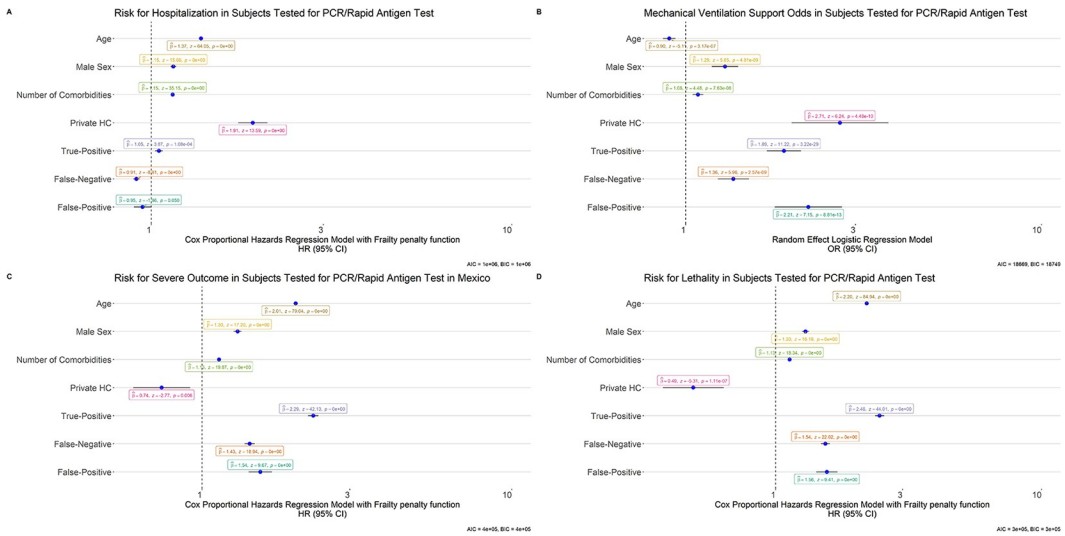

**Fig 3.** Models assessing risk associated to confusion matrix categories in rapid antigen test results compared to qRT-PCR for COVID-19 outcomes including hospitalization (A), requirement for intubation (B), risk of adverse outcomes (C) and lethality (D). **Abbreviations:** qRT-PCR: Reverse transcription polymerase chain reaction; CKD, Chronic Kidney Disease; CVD: cardiovascular disease; COPD: Chronic Obstructive Pulmonary Disease; OR: Odds Ratio; HR: Hazard ratio; 95%CI: 95% Confidence interval; HC: Healthcare.

and false negative results (OR 1.36, 95%CI 1.23–1.51). When assessing the composite of any severe outcome, we observed a higher risk for cases with true positive results (HR 2.28, 95%CI 2.19–2.37), false positive results (HR 1.54, 95%CI 1.41–1.68) and for false negatives (HR 1.42, 95%CI 1.37–1.47). Finally, mortality risk was the highest for cases with true positive results (HR 2.46, 95%CI 2.36–2.56), followed by cases with false positive (HR 1.55, 95%CI 1.42–1.70) and false negative results (HR 1.53, 95%CI 1.48–1.59, **Fig 3**).

**Characterization of SARS-CoV-2 positive cases using Rapid Ag-T.**   Finally, we compared positive cases detected using qRT-PCR and Rapid Ag-T. As expected, positive SARS-CoV-2 cases detected using Rapid Ag-T have a wider spectrum of disease severity with correspondingly lower rates of hospitalization, intubation, mortality and pneumonia. Cases detected using Rapid Ag-T were younger with lower rates of comorbidity and, notably, less median days from symptom onset to evaluation. Amongst cases assessed using Rapid Ag-T, positive SARS-CoV-2 cases had higher risk for hospitalization in older adults, males and subjects with obesity, immunosuppression, CKD, COPD, diabetes or hypertension. For severe COVID-19 and mortality, we identified higher risk in those with CKD, immunosuppression, hypertension, diabetes, males and older adults (**S1 File**).

## Discussion

Here, we performed a real-world large-scale evaluation of Rapid Ag-T for the detection of SARS-CoV-2 at a community-wide level in Mexico. Rapid Ag-T are primarily used in Mexico City for rapid detection of cases to promote self-isolation and prompt initiation of treatment in severe COVID-19 cases. Given the larger availability of Rapid Ag-T, tested cases are younger and have lower rates of comorbidities previously linked to high risk of severe COVID-19, thus leading to lower rates of severe outcomes likely reflective of the true spectrum of SARS-CoV-2 infection in the community [12, 16]. We observed that age, comorbidity, and time from

symptom onset significantly impact the performance of Rapid Ag-T for SARS-CoV-2 and that optimal performance for these tests decreases after 7–10 days from symptom onset. Furthermore, we identified that positive Rapid Ag-T in cases with negative qRT-PCR have higher risks for severe COVID-19 outcomes, indicating potential benefit for the use of Rapid Ag-T in addition to qRT-PCR testing; notably, cases with false negative results in Rapid Ag-T have slightly higher risk of severe COVID-19 outcomes, with the main determinant for false negative status being the time from symptom onset to test assessment. Finally, older patients with negative qRT-PCR had higher odds of a positive Rapid Ag-T, which might call for implementation of sequential testing using Rapid Ag-T after a negative qRT-PCR in older adults with high clinical suspicion. Our results represent the largest evaluation on the usefulness of Rapid Ag-T in a real-world setting as well as on how some common chronic conditions might modify its accuracy in comparison with qRT-PCR tests. With the recent but limited availability of vaccines to prevent symptomatic SARS-CoV-2, consistent prevention of community-level transmission remains paramount to reduce contagions and prevent mortality until an ideal vaccination threshold can be achieved [17]. In this setting, widespread, frequent and repeated use of Rapid Ag-T is preferable given the limited implementation of large-scale qRT-PCR testing for SARS-CoV-2 in Mexico [4, 18].

The use of POCTs is relevant in pandemic settings, where test results can be used to promote self-isolation, adequate treatment allocation and to further contact tracing to reduce rapid dissemination of SARS-CoV-2 [19]. A recent meta-analysis of rapid POCTs for SARS-CoV-2 infection evaluated the use of Rapid Ag-T in different settings, identifying varying values of sensitivity coupled with high specificity, similar to our study [20]. The authors concluded that Rapid Ag-T can be used as a triage to allocate qRT-PCR testing in limited resource settings, which is compatible with our assessment of the clinical implications of false negatives using Rapid Ag-T to detect SARS-CoV-2 infection [21]. Nevertheless, several considerations should be acknowledged as patients with a false negative result could be misclassified and being sent with ambulatory management, increasing the risk of developing severe COVID-19. Our results show that Rapid Ag-T yield a low sensitivity but a very high specificity for detection of SARS-CoV-2 when compared to qRT-PCR. Despite the low sensitivity, the positive and negative predictive values are high likely due to the high prevalence of SARS-CoV-2 in the community [20]. This is consistent with reports of a re-analysis of published data of diagnostic accuracy of qRT-PCR SARS-CoV-2 testing, which described that the risk of false positives increases when extending testing strategies, with increased in false negatives being attributable to local outbreaks [22]. Furthermore, a recent study from Cameroon yields that Rapid-Ag testing is highly correlated and specific to lower cycle threshold values from qRT-PCR testing, providing evidence that Rapid Ag-T could be used as a first approach to evaluate highly transmissible SARS-CoV-2 infection [15]. Lower cycle threshold values correlate with higher viral loads, which in turn may increase risk of severe COVID-19 as suggested by our results. Notably, cases with positive Rapid-Ag testing and negative qRT-PCR were more likely to have >7 days since symptom onset, which may affect detection performance for qRT-PCR testing and also delay access to prompt treatment. In this setting, a positive Rapid-Ag test should be sufficient to allocate treatment and identify cases at the highest risk of complications.

Interestingly, diagnostic performance metrics were very similar to the more controlled setting of the Rapid Ag-T study in Cameroon, reflecting that despite possible differences in testing implementation, results are largely reproducible. In our study, Rapid Ag-T in Mexico City had a higher LR+ compared with the rest of the country, with equally high rates of false negative results using rapid antigen testing; notably, this testing modality is being used in this location to track trends of the COVID-19 pandemic. Although local authorities have promoted the

use of Rapid-Ag T around all the country since November 2020, our results show that there is an unequal testing capacity for each state in Mexico. This inequality in testing could be related to unequal socioeconomic and demographic factors reported during the COVID-19 pandemic rather than individual conditions related to testing performance [23]. Special caution should be taken when evaluating and communicating negative results of rapid antigen tests for SARS-CoV-2, which if misinterpreted could be misleading and reduce adherence for self-isolation of asymptomatic cases identified via contact tracing [24].

Prior to the recommendation from Mexican authorities for the widespread use of Rapid Ag-T, the Mexican Ministry of Health selected cases for qRT-PCR testing based on a sentinel-surveillance system which identified cases based on the presence of respiratory symptoms, leading to an overrepresentation of severe and critical COVID-19 cases [7]. Our group previously profiled cases with non-respiratory symptoms and asymptomatic SARS-CoV-2 infections in Mexico City; we identified that no single symptom offered a reliable assessment of disease severity at the time of initial evaluation, even in population at high risk of contagion such as healthcare workers, as has been confirmed by a recent meta-analysis [25–27]. Given recent evidence which highlights the potential of pre-symptomatic and asymptomatic SARS-CoV-2 transmission and the usefulness of contact tracing as a complement to social distancing and community mitigation policies, SARS-CoV-2 testing should be extended to cases with recent contact with known COVID-19 cases despite the absence of symptoms [24, 28, 29]. Unfortunately, POCTs have relevant limitations on its diagnostic performance which may question its widespread use to inform public policy or for clinical decision making. Particularly, Rapid Ag-T require an active and symptomatic infection and sampling must be done no later than 7 days after beginning of symptoms, while qRT-PCR can be used to assess asymptomatic cases and requires less amount of sample to yield a positive result [30]. Low sensitivity in Rapid Ag-T could also be attributable to additional factors including varying degrees of quality implementation and lack of standardization in testing; however, this should be considered when these tests are implemented at a larger scale, particularly for POCTs. Our data similarly suggests that the time-varying diagnostic performance of Rapid Ag-T might have similar shortcomings to those observed in qRT-PCR testing and which need to be considered when using the result of either method to inform decision-making [31, 32]. Future studies should investigate the utility of Rapid Ag-T as triage for qRT-PCR use in asymptomatic SARS-CoV-2 infection as well as the ideal time frames to reduce the likelihood of discordant results when implementing sequential testing.

Our study had some strengths and limitations. We are using a large national registry of COVID-19 cases, many of whom were tested using both Rapid Ag-T and qRT-PCR in a real-world setting which allowed us to reasonably assess diagnostic performance of Rapid Ag-T to detect SARS-CoV-2 infection. We were also able to assess the clinical impact of discordant results on COVID-19 outcomes as well as predictors which indicate settings where additional testing might be useful to reduce the externalities associated with false negative results. Regarding the limitations to be acknowledged is the potential influence of a spectrum effect, where diagnostic accuracy measures vary according to COVID-19 prevalence and the potential detection bias of only testing most cases once, likely missing infections who were initially categorized as false negative with qRT-PCR early in the course of infection [33, 34]. Furthermore, the use of the sentinel surveillance system to detect and report COVID-19 cases in Mexico likely skews detection towards more severe cases who may also have longer time from symptom onset to evaluation, increasing time-dependent heterogeneity within estimation of predictive accuracy measures [31]. Similarly, since the largest number of Rapid Ag-T were conducted in Mexico City, caution should be taken when generalizing these results to the rest of the country. The lack of disaggregated symptom data prevents assessment of the influence

of various symptom clusters in modifying disease detection with different testing modalities, which remains as an area of opportunity for future research [35]. Another limitation to be acknowledged is that the NESS dataset does not include a variable acknowledging whether there was any delay in time between the performance of both qRT-PCR and Rapid Ag-T testing, which may influence testing performance; however, similarities to other controlled studies on Rapid Ag-T in other settings show similar diagnostic performance [15]. Finally, since the specific Rapid Ag-T were not clearly labeled in the registry, disaggregated diagnostic performance metrics by individual tests could not be estimated.

In conclusion, Rapid Ag-T could be a useful strategy to extend SARS-CoV-2 screening and track trends of the COVID-19 pandemic in Mexico during high transmissibility periods. Rapid Ag-T have poor sensitivity with high specificity but in a setting of local outbreaks, these tests might have high predictive values and be a helpful complement to contact tracing if properly implemented. Rapid Ag-T should be performed widely and frequently to increase the usefulness of low-sensitive tests and increase diagnostic accuracy as well as to guide allocation of qRT-PCR testing in low-resource settings [4, 20, 36]. Our results could inform situations when a discordant result of Rapid Ag-T for SARS-CoV-2 could be expected and the associated clinical implication of using test results for policy and clinical decision making. The use of Rapid Ag-T warrants future evaluations regarding the influence of symptom presentation, recent contact with confirmed COVID-19 case and disease severity on test accuracy and its role in detecting asymptomatic infection as a complement to contact tracing.

## Supporting information

**S1 Checklist.**
(DOCX)

**S1 File.**
(DOCX)

## Acknowledgments

NEAV, ECG, CAFM, AMS, JPBL and AVV are enrolled at the PECEM program of the Faculty of Medicine at UNAM. JPBL, NEAV and AVV are supported by CONACyT. The authors would like to acknowledge the invaluable work of all of Mexico's healthcare community in managing the COVID-19 epidemic. Its participation in the COVID-19 surveillance program has made this work a reality, we are thankful for your effort.

## Author Contributions

**Conceptualization:** Omar Yaxmehen Bello-Chavolla, Neftali Eduardo Antonio-Villa, Luisa Fernández-Chirino, Enrique C. Guerra, Carlos A. Fermín-Martínez, Alejandro Márquez-Salinas, Arsenio Vargas-Vázquez, Jessica Paola Bahena-López.

**Data curation:** Omar Yaxmehen Bello-Chavolla, Neftali Eduardo Antonio-Villa, Luisa Fernández-Chirino, Enrique C. Guerra, Carlos A. Fermín-Martínez, Alejandro Márquez-Salinas, Arsenio Vargas-Vázquez, Jessica Paola Bahena-López.

**Formal analysis:** Omar Yaxmehen Bello-Chavolla, Neftali Eduardo Antonio-Villa, Luisa Fernández-Chirino, Enrique C. Guerra, Carlos A. Fermín-Martínez, Alejandro Márquez-Salinas, Arsenio Vargas-Vázquez, Jessica Paola Bahena-López.

**Funding acquisition:** Omar Yaxmehen Bello-Chavolla.

**Investigation:** Omar Yaxmehen Bello-Chavolla, Neftali Eduardo Antonio-Villa, Luisa Fernández-Chirino, Enrique C. Guerra, Carlos A. Fermín-Martínez, Alejandro Márquez-Salinas, Arsenio Vargas-Vázquez, Jessica Paola Bahena-López.

**Methodology:** Omar Yaxmehen Bello-Chavolla, Neftali Eduardo Antonio-Villa, Luisa Fernández-Chirino, Carlos A. Fermín-Martínez, Alejandro Márquez-Salinas, Arsenio Vargas-Vázquez, Jessica Paola Bahena-López.

**Project administration:** Omar Yaxmehen Bello-Chavolla.

**Resources:** Omar Yaxmehen Bello-Chavolla.

**Software:** Omar Yaxmehen Bello-Chavolla.

**Supervision:** Omar Yaxmehen Bello-Chavolla.

**Validation:** Omar Yaxmehen Bello-Chavolla, Enrique C. Guerra, Jessica Paola Bahena-López.

**Visualization:** Omar Yaxmehen Bello-Chavolla, Neftali Eduardo Antonio-Villa, Luisa Fernández-Chirino, Enrique C. Guerra, Carlos A. Fermín-Martínez, Alejandro Márquez-Salinas, Arsenio Vargas-Vázquez, Jessica Paola Bahena-López.

**Writing – original draft:** Omar Yaxmehen Bello-Chavolla, Neftali Eduardo Antonio-Villa, Luisa Fernández-Chirino, Enrique C. Guerra, Carlos A. Fermín-Martínez, Alejandro Márquez-Salinas, Jessica Paola Bahena-López.

**Writing – review & editing:** Omar Yaxmehen Bello-Chavolla, Neftali Eduardo Antonio-Villa, Luisa Fernández-Chirino, Enrique C. Guerra, Carlos A. Fermín-Martínez, Alejandro Márquez-Salinas, Arsenio Vargas-Vázquez, Jessica Paola Bahena-López.

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
