## [Decision Letter · Decision Letter 0]

29 Apr 2021

PONE-D-21-07222

Diagnostic performance and clinical implications of rapid SARS-CoV-2 antigen testing in Mexico using real-world nationwide COVID-19 registry data

PLOS ONE

Dear Dr. Bello-Chavolla,

Thank you for submitting your manuscript to PLOS ONE. After careful consideration, we feel that it has merit but does not fully meet PLOS ONE’s publication criteria as it currently stands. Therefore, we invite you to submit a revised version of the manuscript that addresses the points raised during the review process.

One of the reviewers commented that a lot of the findings possibly reflect characteristics of individuals receiving antigen tests, rather than test performance. Please consider this issue carefully.

We look forward to receiving your revised manuscript.

Kind regards,

Etsuro Ito

Academic Editor

PLOS ONE

Journal Requirements:

[NEAV, ECG, CAFM, AMS, JPBLand AVVare enrolled at the PECEM program of the Faculty of Medicine at UNAM. JPBL, NEAVand AVV are supported by CONACyT.]

 [The funders had no role in study design, data collection and analysis, decision to publish, or preparation of the manuscript.]

3. We note you have included a table to which you do not refer in the text of your manuscript. Please ensure that you refer to Table 2 in your text; if accepted, production will need this reference to link the reader to the Table.

Reviewers' comments:

Reviewer's Responses to Questions

**Comments to the Author**

1. Is the manuscript technically sound, and do the data support the conclusions?

Reviewer #1: Yes

Reviewer #2: No

2. Has the statistical analysis been performed appropriately and rigorously? 

Reviewer #1: Yes

Reviewer #2: I Don't Know

3. Have the authors made all data underlying the findings in their manuscript fully available?

Reviewer #1: Yes

Reviewer #2: Yes

4. Is the manuscript presented in an intelligible fashion and written in standard English?

Reviewer #1: Yes

Reviewer #2: Yes

5. Review Comments to the Author

Reviewer #1: Research article: PONE-D-21-07222

Title: Diagnostic performance and clinical implications of rapid SARS-CoV-2 antigen testing

in Mexico using real-world nationwide COVID-19 registry data

Bello-Chavolla and colleagues have submitted an interesting and well written article presenting a retrospective study assessing the Diagnostic performance and clinical implications of rapid SARS-CoV-2 antigen testing that was performed in Mexico. Using more than 18,000 patients tests by qRT-PCR and Rapid Antigen tests, they conclude that the Rapid Ag-T could be used as an alternative to qRT-PCR for large scale SARS-CoV-2 testing in Mexico. However they highlight that the interpretation of Rapid Ag-T results should be done with caution in order to minimize the risk associated with what they consider false results based on the qRT-PCR use as reference.

There are however some points to address before being considered for publication in PlosOne

1. The authors should add the actual “optimal” performance of the Rapid Ag-T between 7-10 days after symptoms in the abstract

2. In the line 192 there are mentioned positive and negative results and I was wondering if there were no indeterminate results and what was the process to handle the indeterminate

3. In line 204, the authors mentioned that 18,446 cases had both rapid test and qRT-PCR but it would be great to mention if the samples were collected at the same time and if not what was the delay between the sample collection and the test knowing that it might affect the comparison

4. In line 205 the authors mentioned “the cases” but need to explain who they actually talk about.

5. In line 208 it would be great to the percentage of each group for ease of reading

6. Do the author have more information about the correlation between the rapid test and the Ct Value of the PCR. This is an important information in how to consider the performance of the rapid tests vs PCR

7. In line 274 the author should consider the published algorithm https://www.thelancet.com/pdfs/journals/laninf/PIIS1473-3099(21)00132-8.pdf . The authors should compare their results to field evaluation of rapid tests in asymptomatic done in Cameroon and the suggested algorithm to see what could be proposed to Mexico

Reviewer #2: Evaluating the performance of rapid antigen tests compared to RT-PCR is important for understanding how best to use them; however, I have some serious concerns about the methods and clarity of this paper. While the stated aim of the investigation is to assess the performance of rapid antigen tests for diagnosis of SARS-CoV-2 infection and to examine the clinical implications of the discrepancies in its results compared to qRT-PCR test, to me, a lot of the findings possibly reflect characteristics of individuals receiving antigen tests, rather than test performance. This may be an important finding in itself (i.e. who is getting an antigen test and who gets a positive vs negative result), but it is not clearly framed that way.

Also, it is not clear in the methods whether samples for Rapid Ag-T and qRT-PCR were collected at the same time. This is very important for interpreting test performance and needs to be clarified. If the two samples were not collected at the same time, then differing test results may reflect different stages of infection rather than test performance.

Thirdly, why did some people only receive one test and others receive both? Based on lines 122-125, it sounds like there was likely some bias in who received both (i.e. those who received both tests have a higher pretest probability or based on the facility).This language could be clarified and the bias should be acknowledged.

Fourth, in the mixed-effects Cox regression model, line 173 says the models were adjusted for covariates, but does not specify what covariates. As mentioned above, I am not sure I believe that the performance of Rapid Ag-T predicts hospitalization, mortality, or intubation, but rather it is a reflection of who is receiving the test. There are a lot of potential confounders here.

Please include the name(s) of the qRT-PCR test(s) used in the methods.

The analysis combines results from three antigen tests. While reading the results, I was wondering if test performance could be broken down for each test. In reading the limitations, the authors state that this was not possible. Could the fact that it was not possible be made clearer in the methods?

Some of the results read like the methods were decided based on other results. For example, line 218: "Given the aforementioned observation of time-varying performance according to the time from symptom onset, we used time-dependent ROC curves to model changes in diagnostic performance over time for detection of SARS-Cov-2 infection". First, this concerns me that these decisions were not made apriori. Second, these details belong in the methods rather than the results.

Other comments:

-Line 67: The q in qRT-PCR is not defined.

-Lines 192-193. The authors state "The positivity rate was lower for Rapid Ag-T compared to qRT-PCR", but don't include the positivity rate or any statistical test. I realize now it is in Table 1, but might also be useful to have in the text.

-It looks like a Kappa statistic is presented (line 207), but I could not find any mention of this analysis in the methods.

-For some findings the Hazard Ratios are in the text, while for others they were not (e.g. lines 232-234).

-Line 305: I'm not sure instauration is the correct word here.

Abstract:

-Line 25: I would argue that SARS-CoV-2 testing capacity is more important as a mitigation strategy rather than for monitoring epidemic dynamics.

-Acronyms (qRT-PCR and AUROC) should be spelled out.

-Results: I would like to see the AUROC/HR results in there

Table 1:

-I think 1RT-PCR should be qRT-PCR

-Specify what statistics are reported for age.

-There is an * next to Time to assessment, but no corresponding footnote.

-I could not find a description of how the p-values were calculated in the methods. Is the p-value comparing just qRT-PCR and Rapid antigen test, or all three categories (qRT-PCR, Rapid antigen test, both tests)?

-The font on Figures 2 and 3 is very small, making it difficult to read.

6. PLOS authors have the option to publish the peer review history of their article (what does this mean?). If published, this will include your full peer review and any attached files.

Reviewer #1: **Yes: **Yap Boum

Reviewer #2: No

---

## [Author Response · Author response to Decision Letter 0]

25 Jun 2021

RESPONSE TO REVIEWERS

REVIEWER COMMENTS

REVIEWER 1: 

Bello-Chavolla and colleagues have submitted an interesting and well written article presenting a retrospective study assessing the Diagnostic performance and clinical implications of rapid SARS-CoV-2 antigen testing that was performed in Mexico. Using more than 18,000 patients tests by qRT-PCR and Rapid Antigen tests, they conclude that the Rapid Ag-T could be used as an alternative to qRT-PCR for large scale SARS-CoV-2 testing in Mexico. However, they highlight that the interpretation of Rapid Ag-T results should be done with caution in order to minimize the risk associated with what they consider false results based on the qRT-PCR use as reference.

R) We appreciate the invaluable effort from both reviewers to revise our manuscript. We have considered each comment and included a brief justification for each response wherever appropriate. 

There are however some points to address before being considered for publication in PlosOne

1. The authors should add the actual “optimal” performance of the Rapid Ag-T between 7-10 days after symptoms in the abstract.

R) We have added the performance of Rapid Ag-T between 7-10 days into the abstract section.

2. In the line 192 there are mentioned positive and negative results and I was wondering if there were no indeterminate results and what was the process to handle the indeterminate

R) According to the National Epidemiological Surveillance System (NESS) dictionary dataset, the variable related to the result for each PCR-RT and Rapid-Antigen test is reported in five categories: Positive, negative, pending result, non-adequate sample or case without reported test. For this analysis, we included only positive and negative results, whether the rest of the population was excluded for this analysis. The number of subjects excluded, and the flow-diagram of our selected population is presented in supplementary figure 2, including cases with indeterminate results. We have included a brief mention of these cases in the Results section.

3. In line 204, the authors mentioned that 18,446 cases had both rapid test and qRT-PCR but it would be great to mention if the samples were collected at the same time and if not what was the delay between the sample collection and the test knowing that it might affect the comparison. 

R) Thank you for this observation. The NESS was conducted under the guidance of the “STANDARDIZED GUIDELINES FOR THE EPIDEMIOLOGICAL AND LABORATORY SURVEILLANCE OF VIRAL RESPIRATORY DISEASE” from the Institute of Epidemiological Diagnosis and Reference (InDRE) (https://coronavirus.gob.mx/wp-content/uploads/2021/02/Lineamiento_VE_y_Lab_Enf_Viral_Ene-2021_290121.pdf, version in Spanish). 

According to the last version released on January-2021, all subjects tested with either rapid-antigen or qRT-PCR test should have less than 7 days from any COVID-19 related symptom onset to be considered for testing Furthermore, according to the algorithm proposed by the InDRE to manage nasopharyngeal samples for rapid-antigen test, all negative cases need be re-tested at the same moment with qRT-PCR test to confirm or discharge the diagnosis. Hence, we could expect that both testing methods were sampled at the approximately moment; we have included this specification in the revised Methods section. Nevertheless, the NESS dataset does not report whether any delays occurred for each individual testing method was performed. This has now been mentioned as a limitation in the study in the discussion section. 

4. In line 205 the authors mentioned “the cases” but need to explain who they actually talk about.

R) Thank you for this observation. We referred to subjects tested with qRT-PCR test only. We have modified the following line to: “Subjects tested with qRT-PCR test had higher rates of chronic comorbidities and COVID-19 complications compared to cases with Rapid Ag-T only.”

5. In line 208 it would be great to the percentage of each group for ease of reading

R) We have included the percentage of each classification group in the new version of our manuscript. 

6. Do the author have more information about the correlation between the rapid test and the Ct Value of the PCR. This is an important information in how to consider the performance of the rapid tests vs PCR

R) Although this is an important and relevant question, we do not have the individual CT values for each tested subject recorded in the NESS. According to the “Listed antigen test approved to detect SARS-CoV2 in Mexico (https://www.gob.mx/salud/documentos/listado-de-pruebas-de-antigeno-para-sars-cov-2?state=published, in Spanish)" all commercial kits were validated by qRT-PCR with the lowest threshold accepted (250 copies) according to the InDRE following the RT-PCR Berlin protocol (https://www.eurosurveillance.org/content/10.2807/1560-7917.ES.2020.25.3.2000045). 

We have included a brief paragraph regarding recently published real-word evidence of the correlation of CT values and Ag-Test performed in Cameroon by Boum Yap, et al (https://www.thelancet.com/pdfs/journals/laninf/PIIS1473-3099(21)00132-8.pdf) in the discussion section of our manuscript. 

7. In line 274 the author should consider the published algorithm https://www.thelancet.com/pdfs/journals/laninf/PIIS1473-3099(21)00132-8.pdf . The authors should compare their results to field evaluation of rapid tests in asymptomatic done in Cameroon and the suggested algorithm to see what could be proposed to Mexico.

R) We appreciate this recommendation. The recently published article by Boum Yap, et al yields interestingly results from the use of antigen and antibody rapid diagnostic testing for COVID-19 in Cameroon. Equivalent results are reported within our manuscript related to sensitivity and specificity values. Unfortunately, the NESS does not report the presence of COVID-19 related symptoms within the national dataset. Conversely, the Epidemiological Surveillance System of Mexico City has recently published their COVID-19 population dataset, which included the COVID-19 related symptomatology, which could be a proposal within the approach performed by Boum Yap et al. Nevertheless, the lack of individual CT values, antibody testing and specific testing date posse an important limitation to evaluate the approach proposed by these authors. We have included a brief discussion of the above-mentioned reference within the new discussion section of our manuscript.

REVIEWER #2

Evaluating the performance of rapid antigen tests compared to RT-PCR is important for understanding how best to use them; however, I have some serious concerns about the methods and clarity of this paper. While the stated aim of the investigation is to assess the performance of rapid antigen tests for diagnosis of SARS-CoV-2 infection and to examine the clinical implications of the discrepancies in its results compared to qRT-PCR test, to me, a lot of the findings possibly reflect characteristics of individuals receiving antigen tests, rather than test performance. This may be an important finding in itself (i.e. who is getting an antigen test and who gets a positive vs negative result), but it is not clearly framed that way.

R) We appreciate the suggestions proposed by the reviewer. We agree that there are some methodological limitations that we must acknowledge. 

According to the Institute of Epidemiological Diagnosis and Reference (InDRE), there is a protocol and a standardized guideline for conducting testing for SARS-CoV-2 in Mexico (https://coronavirus.gob.mx/wp-content/uploads/2021/02/Guideline_VE_y_Lab_Enf_Viral_Ene-2021_290121.pdf). The National Epidemiological Surveillance System (NESS) based their evaluations on this protocol and generates the appropriate information to analyze and follow up on suspected cases of COVID-19. This database is of public domain, so clinical research related to the monitoring and management of COVID-19 in Mexico can be verified and generated. However, the use of rapid tests to detect SARS-CoV-2 in Mexico has been strongly promoted by local authorities since October 2020. 

However, InDRE recognizes these limitations and has implemented algorithms for the collection of samples in patients with suspected COVID-19, where it is mentioned that all patients who were tested with a rapid antigen test and had a negative result with strong suspicion of infection based on symptoms or contact tracing information, need to be retested with qPCR-RT to confirm the diagnosis. Although we could not evaluate whether this strategy was completely implemented, we could evaluate the distribution and increasing testing capacity in each state in Mexico, rather than individual factors related to testing discrepancies. These issues have been now discussed in the limitations section of our revised manuscript.

1. Also, it is not clear in the methods whether samples for Rapid Ag-T and qRT-PCR were collected at the same time. This is very important for interpreting test performance and needs to be clarified. If the two samples were not collected at the same time, then differing test results may reflect different stages of infection rather than test performance.

R) As mentioned above and by the previous reviewer, the NESS and the InDRE clarified in the “STANDARDIZED GUIDELINES FOR THE EPIDEMIOLOGICAL AND LABORATORY SURVEILLANCE OF VIRAL RESPIRATORY DISEASE” from the Institute of Epidemiological Diagnosis and Reference (InDRE) (https://coronavirus.gob.mx/wp-content/uploads/2021/02/Lineamiento_VE_y_Lab_Enf_Viral_Ene-2021_290121.pdf, version in Spanish) that testing of both qRT-PCR and Rapid Ag-T was performed in the same day of assessment. Furthermore, all subjects tested with either rapid-antigen or qRT-PCR test should have less than 7 days from any COVID-19 related symptom onset to be considered as viable cases. This is further specified in the new version of our manuscript. 

2. Thirdly, why did some people only receive one test and others receive both? Based on lines 122-125, it sounds like there was likely some bias in who received both (i.e. those who received both tests have a higher pretest probability or based on the facility). This language could be clarified and the bias should be acknowledged.

R) This represents a clear temporal bias inherent to the registry and the database, which is discussed in our manuscript as rapid antigen testing was only implemented during early November in 2020. Additionally, the use of rapid tests is currently focalized on urbanized regions, which represents inequality sampling related to socioeconomical factors in Mexico that has already been previously reported by our study group (https://www.ncbi.nlm.nih.gov/pmc/articles/PMC7337730). Cases with both tests did have higher rates of comorbidities and were older compared to cases with only one test and pre-test probability may have influenced the allocation of both tests; this occurs in accordance with the algorithm proposed by Boum Yap et al. and is expected as testing is currently implemented with a limited capacity in Mexico. These issues recognized as a limitation that could reflect social and demographic factors (e.g., adequate distribution of tests in Mexico, economic accessibility of each subject, report of each evaluation center) that should be further evaluated in our country. 

3. Fourth, in the mixed-effects Cox regression model, line 173 says the models were adjusted for covariates, but does not specify what covariates. As mentioned above, I am not sure I believe that the performance of Rapid Ag-T predicts hospitalization, mortality, or intubation, but rather it is a reflection of who is receiving the test. There are a lot of potential confounders here.

R) Thank you for this observation. We have included the covariates that were used for our analysis of each outcome, these include age, sex, diabetes, arterial hypertension, CORP, immunosuppression, cardiovascular disease, obesity, asthma and chronic kidney disease. Although, we have included social and clinical related factors that have been demonstrated that could alter the relationship within each outcome, we always need to consider a residual confounding effect regarding non-measured variables, particularity related to sociodemographic related factors which are not assessed within the NESS dataset. We acknowledge this limitation and included a brief paragraph of this response within the new version of our manuscript.

4. Please include the name(s) of the qRT-PCR test(s) used in the methods.

R) According to the Mexican Health Secretary, up to 6th of April of 2021, there were 85 approved qRT-PCR testing kits for SARS-CoV-2 in Mexico. Each one had its own CT threshold which could be consulted within the following URL: https://www.gob.mx/salud/documentos/listado-de-pruebas-moleculares-por-rt-pcr-monoplexado-sars-cov-2?state=published (Spanish version). A brief paragraph regarding all available qRT-PCR test(s) used in the NESS is presented in the supplementary methods section.

The analysis combines results from three antigen tests. While reading the results, I was wondering if test performance could be broken down for each test. In reading the limitations, the authors state that this was not possible. Could the fact that it was not possible be made clearer in the methods?

R) We have included a brief statement regarding this limitation in the new version of our manuscript. 

Some of the results read like the methods were decided based on other results. For example, line 218: "Given the aforementioned observation of time-varying performance according to the time from symptom onset, we used time-dependent ROC curves to model changes in diagnostic performance over time for detection of SARS-Cov-2 infection". First, this concerns me that these decisions were not made apriori. Second, these details belong in the methods rather than the results.

R) This statement and analysis was performed as a sensitivity analysis that could further strengthen our conclusions. We have included a brief statement regarding the justification of using time-dependent ROC curves in the methods section.

Other comments:

-Line 67: The q in qRT-PCR is not defined.

R) We have added the complete abbreviation of qRT-PCT (quantitative reverse transcription polymerase chain). 

-Lines 192-193. The authors state "The positivity rate was lower for Rapid Ag-T compared to qRT-PCR", but don't include the positivity rate or any statistical test. I realize now it is in Table 1, but might also be useful to have in the text.

R) We have included the positivity rate for each group in the revised version of our manuscript.

-It looks like a Kappa statistic is presented (line 207), but I could not find any mention of this analysis in the methods.

R) We have included the following line within the methods section: We estimated the concordance of both testing methods using Cohen’s Kappa coefficient.

-For some findings the Hazard Ratios are in the text, while for others they were not (e.g. lines 232-234).

R) For better consistency of presentation of our results, we have omitted the punctual HR parameter as they are presented in full in Figure 2. 

-Line 305: I'm not sure instauration is the correct word here.

R) We have modified this line for: “Prior to the recommendation from Mexican authorities for the widespread use of Rapid Ag-T, the Mexican Ministry of Health selected cases for qRT-PCR testing based on a sentinel-surveillance system”

Abstract:

-Line 25: I would argue that SARS-CoV-2 testing capacity is more important as a mitigation strategy rather than for monitoring epidemic dynamics.

R) We have included both acceleration as both reflect the complexity of massive testing to monitor the COVID-19 epidemic. We have changed the following sentence to: “SARS-CoV-2 testing capacity is important to monitor epidemic dynamics and as a mitigation strategy.”

-Acronyms (qRT-PCR and AUROC) should be spelled out.

R) We have included this in the abstract section. 

-Results: I would like to see the AUROC/HR results in there

R) We have made this change.

Table 1:

I think 1RT-PCR should be qRT-PCR

R) We have corrected this typo in the new version of our manuscript.

-Specify what statistics are reported for age.

R) Age is presented in mean and standard deviation. This is specified for all our tables’ footnotes.

-There is an * next to Time to assessment, but no corresponding footnote.

R) This “*” stands for the definition of time to assessment as the time since COVID-19 related symptoms onset up to the registration of the tested subject in the medical unit.

-I could not find a description of how the p-values were calculated in the methods. Is the p-value comparing just qRT-PCR and Rapid antigen test, or all three categories (qRT-PCR, Rapid antigen test, both tests)?

R) In table 1, we performed a global comparison of all three groups using ANOVA, Kruskal-Wallis, or Chi-Square teste wherever appropriate. We have specified this as a footnote.

-The font on Figures 2 and 3 is very small, making it difficult to read.

R) We have resized both figures in the revised version of our manuscript.

---

## [Decision Letter · Decision Letter 1]

26 Jul 2021

PONE-D-21-07222R1

Diagnostic performance and clinical implications of rapid SARS-CoV-2 antigen testing in Mexico using real-world nationwide COVID-19 registry data

PLOS ONE

Dear Dr. Bello-Chavolla,

Thank you for submitting your manuscript to PLOS ONE. After careful consideration, we feel that it has merit but does not fully meet PLOS ONE’s publication criteria as it currently stands. Therefore, we invite you to submit a revised version of the manuscript that addresses the points raised during the review process.

The comments from one reviewer seem minor. Please revise your manuscript carefully.

We look forward to receiving your revised manuscript.

Kind regards,

Etsuro Ito

Academic Editor

PLOS ONE

Journal Requirements:

Reviewers' comments:

Reviewer's Responses to Questions

**Comments to the Author**

1. If the authors have adequately addressed your comments raised in a previous round of review and you feel that this manuscript is now acceptable for publication, you may indicate that here to bypass the “Comments to the Author” section, enter your conflict of interest statement in the “Confidential to Editor” section, and submit your "Accept" recommendation.

Reviewer #1: All comments have been addressed

Reviewer #2: (No Response)

2. Is the manuscript technically sound, and do the data support the conclusions?

Reviewer #1: Yes

Reviewer #2: Yes

3. Has the statistical analysis been performed appropriately and rigorously? 

Reviewer #1: Yes

Reviewer #2: Yes

4. Have the authors made all data underlying the findings in their manuscript fully available?

Reviewer #1: Yes

Reviewer #2: Yes

5. Is the manuscript presented in an intelligible fashion and written in standard English?

Reviewer #1: Yes

Reviewer #2: Yes

6. Review Comments to the Author

Reviewer #1: the authors have well adressed the reviewer's comments. This article will surely contribute to the global roll out of Rapid antigenic tests that are critical for the fight againts COVID-19.

Reviewer #2: Thank you for your responses and revisions. I have a few remaining comments and I apologize if I missed these the first time around.

Line 111-112: What does 'used extensively for tracking the epidemic' mean? Does this mean that Rapid Ag-T tests were used other than for testing suspected COVID-19 cases or persons with epidemiological association with a suspected case? Is 'epidemiological association' synonymous with 'close contact'?

Lines 113-117: It sounds like you are defining a 'false positive' result as a confirmed SARS-CoV-2 infection. Is this correct? Could you further clarify this in your methods?

Line 117-119: Thank you for adding in this detail, this clarifies the process for negative Rapid Ag-T individuals. Do you have any information on if there was a delay if a person was Rapid Ag-T positive, qRT-PCR negative? This relates to my comment below

Line 243: I am very surprised by the finding that the risk of intubation was higher for false positive test results. Can you further explain why this may be in your discussion? If you believe these are confirmed SARS-CoV-2 infections, why do you hypothesize that the qRT-PCR was negative?

7. PLOS authors have the option to publish the peer review history of their article (what does this mean?). If published, this will include your full peer review and any attached files.

Reviewer #1: **Yes: **Yap Boum

Reviewer #2: No

---

## [Author Response · Author response to Decision Letter 1]

4 Aug 2021

RESPONSE TO REVIEWERS

REVIEWER COMMENTS

REVIEWER #1

• The authors have well adressed the reviewer's comments. This article will surely contribute to the global roll out of Rapid antigenic tests that are critical for the fight against COVID-19.

R= Thank you for your comments. We also believe that rapid-Ag testing is a fundamental strategy for epidemiological surveillance of the COVID-19 pandemic in Mexico.

REVIEWER #2

Thank you for your responses and revisions. I have a few remaining comments and I apologize if I missed these the first time around.

R= Thank you for your revisions and comments, they have helped us improve our work.

• Line 111-112: What does 'used extensively for tracking the epidemic' mean? Does this mean that Rapid Ag-T tests were used other than for testing suspected COVID-19 cases or persons with epidemiological association with a suspected case? Is 'epidemiological association' synonymous with 'close contact'?

R= Rapid antigen testing has been quicky implemented as the main diagnostic method of SARS-CoV2 infection in Mexico; notably, most COVID-19 testing in Mexico City have been based on this method. To better reflect this, we corrected line 112 as: “These Rapid Ag-T are available in healthcare community-level locations for testing of suspected COVID-19 cases or subjects traced by epidemiological association with a suspected case and they are used extensively for monitoring and tracking COVID-19 incidence rates in Mexico City”

• Lines 113-117: It sounds like you are defining a 'false positive' result as a confirmed SARS-CoV-2 infection. Is this correct? Could you further clarify this in your methods?

R= The definition of “false positive result” was the combination of a positive rapid-ag test and a negative qRT-PCR. Furthermore, all cases with negative Ag-T, but with high suspicion of COVID-19 were eligible for qRT-PCR testing. We further clarify this within the next paragraph in line 115: “Cases with negative Rapid Ag-T but with recently close contact with a confirmed SARS-CoV-2 case and/or compatible clinical symptoms of COVID-19 were eligible for further evaluation with qRT-PCR testing within testing facilities”.

• Line 117-119: Thank you for adding in this detail, this clarifies the process for negative Rapid Ag-T individuals. Do you have any information on if there was a delay if a person was Rapid Ag-T positive, qRT-PCR negative? This relates to my comment below

R= Thank you for this comment. Unfortunately, we don’t have the requested information. This is a limitation that we have acknowledged in our manuscript in line 369: “Another limitation to be acknowledged is that the NESS dataset does not include a variable acknowledging whether there was any delay in time between the performance of both qRT-PCR and Rapid Ag-T testing, which may influence testing performance; however, similarities to other controlled studies on Rapid Ag-T in other settings show similar diagnostic performance”

• Line 243: I am very surprised by the finding that the risk of intubation was higher for false positive test results. Can you further explain why this may be in your discussion? If you believe these are confirmed SARS-CoV-2 infections, why do you hypothesize that the qRT-PCR was negative?

R= Rapid antigen testing has been shown to vary in sensitivity depending on SARS-CoV-2 viral loads. The higher viral load detected by a rapid antigen test may pose a higher risk of adverse COVID-19 outcomes, as shown in our results. As shown in Figure 2, false negative results were most likely to be younger but with >7 days from symptom onset to testing, which may reduce sensitivity of qRT-PCR. This delay in case confirmation may also delay access to care, increasing risk of severe COVID-19 and requirement for invasive ventilation. We discussed this in lines 305-310 as “Lower cycle threshold values correlate with higher viral loads, which in turn may increase risk of severe COVID-19 as suggested by our results. Notably, cases with positive Rapid-Ag testing and negative qRT-PCR were more likely to have >7 days since symptom onset, which may affect detection performance for qRT-PCR testing and delay access to prompt treatment. In this setting, a positive Rapid-Ag test should be sufficient to allocate treatment and identify cases at the highest risk of complications.”

---

## [Editor Report · Decision Letter 2]

9 Aug 2021

Diagnostic performance and clinical implications of rapid SARS-CoV-2 antigen testing in Mexico using real-world nationwide COVID-19 registry data

PONE-D-21-07222R2

Dear Dr. Bello-Chavolla,

We’re pleased to inform you that your manuscript has been judged scientifically suitable for publication and will be formally accepted for publication once it meets all outstanding technical requirements.

Kind regards,

Etsuro Ito

Academic Editor

PLOS ONE

---

## [Editor Report · Acceptance letter]

11 Aug 2021

PONE-D-21-07222R2 

Diagnostic performance and clinical implications of rapid SARS-CoV-2 antigen testing in Mexico using real-world nationwide COVID-19 registry data 

Dear Dr. Bello-Chavolla:

I'm pleased to inform you that your manuscript has been deemed suitable for publication in PLOS ONE. Congratulations! Your manuscript is now with our production department. 

Kind regards, 

on behalf of

Prof. Etsuro Ito 

Academic Editor

PLOS ONE